# Reliability of online, remote neuropsychological assessment in people with and without subjective cognitive decline

Katie A. Peterson[1], Adrian Leddy[1], Michael Hornberger[2]*

**1** Department of Clinical Psychology and Psychological Therapies, Norwich Medical School, University of East Anglia, Norwich, United Kingdom, **2** Department of Medicine, Norwich Medical School, University of East Anglia, Norwich, United Kingdom

* M.Hornberger@uea.ac.uk

## Abstract

Online, remote neuropsychological assessment paradigms may offer a cost-effective alternative to in-person assessment for people who experience subjective cognitive decline (SCD). However, it is vital to establish the psychometric properties of such paradigms. The present study (i) evaluates test-retest reliability of remote, online neuropsychological tests from the NeurOn software platform in people with and without SCD (Non-SCD) recruited from the general population; and (ii) investigates potential group differences in baseline performance and longitudinal change. Ninety-nine participants (SCD N = 44, Non-SCD N = 55) completed seven tests from the NeurOn battery, covering visual and verbal memory, working memory, attention and psychomotor speed. Sixty-nine participants (SCD N = 34, Non-SCD N = 35) repeated the assessment six (+/- one) months later. SCD was classified using the Cognitive Change Index questionnaire. Test-retest reliability of the NeurOn test outcome measures ranged from poor to good, with the strongest evidence of reliability shown for the Sustained Attention to Response Test and Picture Recognition. The SCD group was significantly older than the Non-SCD group so group differences were investigated using analysis of covariance whilst controlling for the effect of age. SCD scored significantly better than Non-SCD for Digit Span Backwards (maximum sequence length) and Picture Recognition (recall of object position) at baseline. However, these were not significant when using the Bonferroni-adjusted alpha level. There were no differences between SCD and Non-SCD in longitudinal change. The results suggest online, remote neuropsychological assessment is a promising option for assessing and monitoring SCD, however more research is needed to determine the most suitable tests in terms of reliability and sensitivity to SCD.

## Author summary

A considerable proportion of the older adult population experiences subjective decline in their thinking skills even though they score within 'normal' limits on screening tests for mild cognitive impairment or dementia. Research suggests that, for a small percentage of

**Data availability statement:** The data that support the findings of the current research are available at the Centre for Open Science Open Science Framework online platform at the following url: https://osf.io/cwskh

**Funding:** The author(s) received no specific funding for this work.

**Competing interests:** I have read the journal's policy and the authors of this manuscript have the following competing interests: Professor Michael Hornberger is the research director at NeurOn. All other authors have declared that no competing interests exist.

these people, their experience of a decline in their thinking skills might indicate an early stage of dementia. It is important for research to identify the earliest markers of dementia as this is when treatments may be most effective. By harnessing computing technology to improve on the accuracy and availability of cognitive assessments, we may be able to identify early and subtle cognitive changes caused by dementia. This study investigated whether online and remote cognitive assessment is a reliable method to assess and monitor thinking skills in the general older adult population. We were able to identify tasks which showed the best evidence for reliability when completed online and remotely by people with and without a subjective experience of cognitive decline, and therefore may be appropriate for monitoring thinking skills in people who are concerned about their cognitive ability. Our findings suggest online cognitive assessment may be a useful and cost-effective alternative to in-person clinic-based assessment, although care should be taken to determine reliability and sensitivity of tests used.

## Introduction

Cognitive and functional impairment associated with dementia places a significant burden on healthcare. This is projected to rise in line with the ageing population in the United Kingdom [1]. Research into treatments has been hampered by the lack of biomarkers for early or pre-symptomatic detection of neurodegenerative disease [2]. Pathophysiological changes of neurodegenerative disease occur years before symptom onset [3–5]. Therefore, earlier detection of dementia is a key priority for research as this is when disease-modifying treatments may be most effective [6,7]. There is emerging evidence that subtle cognitive changes are detectable years before diagnosis in sporadic neurodegenerative disease [6,8].

Neuropsychological assessment is a key tool for the detection and monitoring of cognitive impairment associated with dementia [9]. Better assessment methods are required to detect subtle cognitive changes in early disease stages [10]. The ability to harness advances in technology to collect more comprehensive and frequent data is a key area of interest in dementia research, including the use of digital methods for in-home monitoring of cognition [11]. Unsupervised, online neuropsychological assessment has the potential to increase the availability and frequency of cognitive assessments in order to detect and track subtle changes in cognitive ability [12].

It has been suggested that subjective cognitive decline (SCD) might be an early marker of cognitive impairment due to neurodegeneration [13]. SCD is the self-perception of a decline in cognitive performance despite unimpaired performance on standardised tests sensitive to mild cognitive impairment (MCI) or dementia [14]. Most people with SCD do not progress to MCI or dementia. However, research suggests they are at increased risk of doing so compared to people without SCD [14–16]. Specific factors have been identified to be associated with an increased risk of cognitive decline in people who experience SCD (known as the "SCD plus" criteria): subjective decline in memory, onset within the last five years, onset at age 60+, persistence of SCD, presentation at a memory clinic, and informant-reported cognitive decline [14,15]. A recent meta-analysis identified additional risk factors for objective cognitive decline in people with SCD beyond the SCD plus criteria [17], including biomarkers of Alzheimer's disease pathology (e.g., high amyloid β/ high total tau protein in the brain and/or hippocampal atrophy), the presence of apolipoprotein E4 genotype, comorbid depression or anxiety, smoking status, fewer years of education, and poorer performance on a measure of executive functioning (investigated using Trail-Making Test B [18] performance).

Given that most individuals with SCD will not progress to MCI, it is not recommended to monitor everyone. However, for those with additional risk factors, remote, online neuropsychology offers a low-cost method to assess and monitor cognition over time. Further, given the projected increase in the average age of people living in rural areas in England [1] remote assessment options offer a practical method to support accessibility of neuropsychological assessment services for rural populations [19]. Although such research is in its infancy, initial evidence suggests online neuropsychological assessment, completed remotely, can detect subtle deficits in cognition in people with SCD [20], therefore suggesting that this is a promising tool for the assessment and monitoring of SCD.

It is unclear whether online neuropsychological tests, completed remotely and unsupervised, show acceptable psychometric properties for monitoring cognition. Various factors associated with online, remote test completion may impact on the reliability of results, such as technical issues, computer skills, cognitive and physical abilities affecting computer use, and a lack of supervision and additional instruction [21], meaning that equivalence to in-person tests cannot be assumed. A number of online neuropsychological assessment batteries have been developed which have shown low to high validity and reliability [22–27]. However, there is heterogeneity between the studies in terms of study populations and methods used. Therefore, more data is needed in different populations particularly for online, remote neuropsychological assessment to inform its use in clinical practice [21,28]. The present study evaluates the reliability of remote, online neuropsychological tests, completed without supervision by people with and without SCD recruited online from the general population.

The primary objective of the study is to establish the test-retest reliability of online tests from the NeurOn software platform in people with and without SCD. A selection of NeurOn tests were previously found to have moderate test-retest reliability in healthy older adults and feasibility for completing remotely [27]. The secondary objective of the study is to characterise online neuropsychological test performance in people with and without SCD by investigating group differences in baseline performance and baseline-to-follow-up change. These objectives were achieved.

## Hypotheses

We predicted that:

1. Online neuropsychological tests will show moderate test-retest reliability, in keeping with previous findings [27].

2. People with SCD will show subtle impairment in online, remote neuropsychological test performance compared to people without SCD (Non-SCD), in line with previous research [20].

## Materials and methods

### Ethical approval

Ethical approval was obtained from the University of East Anglia Faculty of Medicine and Health Sciences Research Ethics Subcommittee (ETH2223-0113). All participants provided informed consent electronically via an online consent form.

### The Mantal and NeurOn software platforms

The Mantal software platform (https://mantal.co.uk/) was developed to facilitate the management of online clinical research studies. The NeurOn platform (https://neuropsychology.online/) is an online neuropsychological testing platform for clinicians and researchers. The

NeurOn platform currently contains cognitive tests covering domains including memory, language, visuospatial ability, executive functioning and attention. The tests feature randomised stimulus sets to allow longitudinal cognitive testing with minimal test-retest effects. NeurOn tests can be accessed within the Mantal software platform via an application programming interface. Therefore, participants are only required to create an account with one platform (Mantal) where they can then complete the relevant cognitive tests, pre-selected by the research team.

Test-retest reliability has been evaluated for a selection of the NeurOn tests (Reaction Time, a Go-No/Go test and two subtests of the Virtual Supermarket Task) in a healthy control group who completed the online tests in-person (baseline) and remotely (follow-up), one week apart [27]. The four subtests showed moderate test-retest reliability. In the present analysis, we extended these findings by assessing test-retest reliability for a larger selection of NeurOn tests relevant to SCD in SCD and Non-SCD groups, separately, and for fully remote participation.

## Participants

Participants were included if they met the following eligibility criteria:

**Inclusion criteria.**

- Age 60+ in line with the World Health Organisation definition of old age

- Capacity to give informed consent

- Sufficient computer literacy to complete the online Consent Form

- Fluent in English

- Access to a device (computer or laptop) for the completion of the study

**Exclusion criteria.**

- A diagnosis of a neurological or neurodegenerative condition

- A diagnosis of mild cognitive impairment

- Being under the care of a secondary mental health service, due to the link between severe psychiatric disorders (and some pharmacological treatments) with cognitive dysfunction [29].

We aimed to recruit a sample size of 50 people per group (SCD; Non-SCD) based on similar studies of normative neuropsychological test data [30,31]. Longitudinal research studies with older adults have reported drop-out rates of between 5-37% [32,33]. Therefore, we aimed to recruit 120 participants to factor in an attrition rate in this region (assuming roughly 20%).

## Recruitment

Recruitment began in April 2023. Participants were recruited via advertisement on social media, within the University of East Anglia campus, and via the National Institute for Health Research "Join Dementia Research" register (http://www.joindementiaresearch.nihr.ac.uk) in Norwich.

## Procedure

The Participant Information Sheet was sent to potential participants via email along with a link to the study, hosted on the Mantal clinical research software platform. Potential participants were advised they could take as much time as they like to consider the information

sheet. People who decided to take part in the study were able to register with the study website (using their email address) and complete an online consent form. After completing the consent form, participants were able to access an online eligibility screen which they were asked to complete by indicating whether they met each of the eligibility criteria via check boxes. If participants met all eligibility criteria they were then able to access the full baseline study session. Participants were instructed to use a laptop or desktop computer to complete the study as some of the current versions of the NeurOn tasks do not function correctly if the screen size is too small.

At the baseline session, participants provided demographic information before completing the study measures (mood questionnaires, SCD questionnaire, and NeurOn tests). The following demographic data were collected: age, sex, level of education (1 = did not complete GCSE, 2 = GCSE or equivalent, 3 = A Level or equivalent, 4 = Undergraduate degree or equivalent, 5 = Master's degree or equivalent, 6 = Doctoral degree), self-rated confidence using computers (1= not at all confident; 5 = very confident) since computer literacy may be related to online cognitive test performance [34], self-estimated average sleep time, social interaction (measured using the Duke Social Support Index [35], Social Interaction subscale: max score = 12, with higher scores indicating greater social interaction), previous COVID-19 infection or long-covid since a previous infection has been shown to affect cognition [36], occupation, first part of postcode (as a proxy measure of socioeconomic status) and whether participants had a diagnosis of dyslexia. First part of postcode was converted to a socioeconomic status score using the Indices of Multiple Deprivation produced by the Ministry of Housing, Communities and Local Government [37] to derive an income deprivation percentage for the relevant local authority. Higher scores indicate greater levels of deprivation in the local authority area.

Participants were contacted by email five months after completing their baseline session to invite them to complete their six- (+/- one) month follow up session. Participants repeated the mood questionnaires and the NeurOn tests at follow up. A six-month delay between baseline and follow-up was chosen to be consistent with typical assessment schedules used to monitor cognitive changes in clinical practice (e.g., [38]).

## Measures

Participants completed the following measures online via the Mantal study website:

**Assessment of subjective cognitive decline.** Given the recruitment method precluded detailed screening of participants, we used a validated questionnaire to assess SCD, the 20-item Cognitive Change Index (CCI) [39]. The CCI was developed to assess cognitive complaints in older adults. We defined SCD as a score of 20 or above on the first 12 items of the CCI in accordance with recommendations by the developers of the measure [40]. Participants completed the CCI during the baseline session.

**Mood questionnaires.** Mood was assessed since there are well documented links between mood and cognitive performance [41]. The 15-item version of the Geriatric Depression Scale (GDS-15; [42]) was used to screen for depression. The maximum score is 15. A score of five or above indicates mild depression symptoms; a score of nine or above indicates moderate depression symptoms. The Geriatric Anxiety Inventory (GAI) [43] was used to screen for anxiety. The maximum score is 20. A score of nine or above indicates clinically significant anxiety symptoms. These scales were chosen as they were developed for use in older adult populations, therefore avoiding misattributing signs of normal ageing to depression or anxiety, and are well validated and commonly used.

**Online neuropsychological assessment.** Participants completed computerised neuropsychological tests from the NeurOn software platform within their Mantal account

via an application programming interface within the Mantal study website. The tests can be completed using either touch screen or keyboard input, depending on the capabilities of the equipment used by participants. Participants completed the following tests in the order shown:

1. Picture Encoding: a stimulus encoding phase in which everyday objects are presented on screen at varying locations (top, bottom, left or right). Participants are instructed to remember the pictures and where on the screen they were presented.

2. Simple Reaction Time: participants are instructed to respond to repeated, on-screen stimuli as fast as they can.

3. Digit Span backwards (working memory): participants are required to remember a sequence of digits which are presented one by one on the screen. They must recall the digits in reverse order. The length of the sequence increases until two trials of a sequence length are failed, ending the test.

4. Picture Recognition (visual memory): a recognition phase in which everyday objects (made up of a mixture of previously presented objects during the Picture Encoding phase, and novel objects) are presented on screen. For each item, participants must indicate whether they saw the object before. If they answer 'yes', they are then asked where on the screen the object was presented.

5. Word Encoding: a stimulus encoding phase in which a series of high-frequency words are presented on screen at varying locations (top, bottom, left or right). Participants are instructed to remember the words and where on the screen they were presented.

6. Sustained Attention to Response Test (attention): participants are presented with a series of digits and are instructed to respond to each digit apart from one (the 'no-go' target stimulus). There are 255 trials in the test, therefore requiring sustained attention over time. The task records reaction time, and will identify responses that are "too soon" or anticipatory (i.e., indicating responses that are faster than would be possible if following the rules of the task).

7. Word Recognition (verbal memory): a recognition phase in which a series of words (made up of a mixture of previously presented words during the Word Encoding phase, and novel words) are presented on screen. For each item, participants must indicate whether they saw the word before. If they answer 'yes', they are then asked where on the screen the word was presented.

8. Trail-Making Tests A and B (psychomotor speed, attention): participants are required to click 25 symbols in a certain order as fast as possible. For Trail-Making Test A participants must click numbered circles in order from smallest to largest, whereas for Trail-Making Test B they must alternate between numbers and letters in ascending order.

These tests were selected as they measure cognitive abilities commonly affected in early stages of dementia [44–46].

There was a delay of approximately 10 minutes between the picture/word encoding and recognition subtasks. The full neuropsychological test battery took approximately 20-30 minutes to complete. While the neuropsychological test battery was required to be completed in one sitting, participants were informed they could complete the neuropsychological tests and the questionnaires in separate sittings.

## Analysis

The study used a longitudinal observational case-control design. Participants were grouped (SCD; Non-SCD) according to their score on the CCI. Test-retest reliability of the online neuropsychological tests was assessed in both groups, separately. Performance on the online neuropsychological tests at baseline and the change over time was compared between the two groups. We used an opportunistic, community-based recruitment approach as this is in line with the clinical question, therefore, SCD and Non-SCD groups were not matched for age during recruitment. The selected outcome measures for each cognitive test are detailed in Table 1.

Test-retest reliability was assessed using two-way mixed effects intraclass correlation coefficients (ICC) with absolute agreement as is recommended [47]. Koo and Li [48] suggest the following interpretation of ICC values: less than 0.5 indicates poor reliability, 0.5-0.75 indicates moderate reliability, 0.75-0.9 indicates good reliability, and greater than 0.9 indicates excellent reliability.Chi-square test was conducted to investigate differences in sex, previous COVID-19 infection, long-covid prevalence, and dyslexia prevalence between the two groups. Continuous demographic data were assessed for normality using the Shapiro-Wilk test. The assumption of normality was violated for all continuous demographic measures. Therefore, Mann-Whitney U test was used to test for group differences (SCD versus Non-SCD) in these variables and group statistics reported using median and interquartile range. Analysis of covariance (ANCOVA) was used to explore group differences in baseline and baseline-to-follow-up change scores for each neuropsychological test outcome measure while controlling

**Table 1. Outcome measures for each NeurOn test.**

| NeurOn Test | Outcome measure | Direction of better performance |
|---|---|---|
| *Digit Span Backwards* | **N correct** = total number of correct sequences (max = 16) | ↑ |
| | **N errors** = total number of incorrect sequences | ↓ |
| | **Max length** = maximum sequence length correctly recalled (max = 9) | ↑ |
| *Picture Recognition* | **N correct** = total number of correctly recognised pictures and correct rejections of novel pictures (max = 30) | ↑ |
| | **N position correct** = total number of trials where the position of a recognised picture was correctly identified (max = 15) | ↑ |
| | **False alarms** = total number of 'false alarms', i.e., incorrect recognition of a novel picture | ↓ |
| *Simple Reaction Time* | **Average reaction speed** = mean reaction speed across correct trials (i.e., excluding incorrect trials) | ↓ |
| | **N errors** = total number of incorrect trials | ↓ |
| *Sustained Attention to Response Test* | **N correct** = total number of correct trials | ↑ |
| | **Average reaction speed** = mean reaction speed across correct trials (i.e., excluding incorrect trials) | ↓ |
| | **N errors** = total number of anticipatory and "too soon" responses | ↓ |
| *Trail-Making Test A* | **N errors** = total number of incorrect responses | ↓ |
| | **Time to complete** = total time taken to complete the task | ↓ |
| *Trail-Making Test B* | **N errors** = total number of incorrect responses | ↓ |
| | **Time to complete** = total time taken to complete the task | ↓ |
| *Word Recognition* | **N correct** = total number of correctly recognised words and correct rejections of novel words (max = 30) | ↑ |
| | **N position correct** = total number of trials where the position of a recognised word was correctly identified (max = 15) | ↑ |
| | **False alarms** = total number of 'false alarms', i.e., incorrect recognition of a novel word | ↓ |

for the effect of age. Omega squared ($\omega^2$) was used as a measure of effect size as it is less biased than other effect size measures in small samples [49]. Given each set of ANCOVAs examined 18 dependent variables (NeurOn test outcome measures), a Bonferroni-adjusted alpha level of 0.05/18 = 0.003 was used for the ANCOVA results. Change scores were calculated by subtracting baseline scores from follow-up scores. Data analysis was conducted using JASP (version 0.18.3) [50], R (version 4.0.2) and RStudio (version 2023.12.1) [51].

## Results

### Participant demographics

Fig 1 shows participation and completion rates for each part of the study. Twelve people registered and provided consent to participate but then did not complete the eligibility screen. Therefore, it is presumed they did not meet eligibility for the study. Two people did not complete the CCI and therefore were excluded from the group comparisons. 108 people (SCD N=47, Non-SCD N=61) completed the CCI and at least the study questionnaires at baseline. The demographics and questionnaire scores of the 108 participants are summarised in Table 2. All participants lived in the United Kingdom. The SCD group were significantly older than the Non-SCD group and scored significantly higher for depression and anxiety, however, the medians were well below the clinical range for both tests. As expected, CCI score was significantly higher in the SCD group.

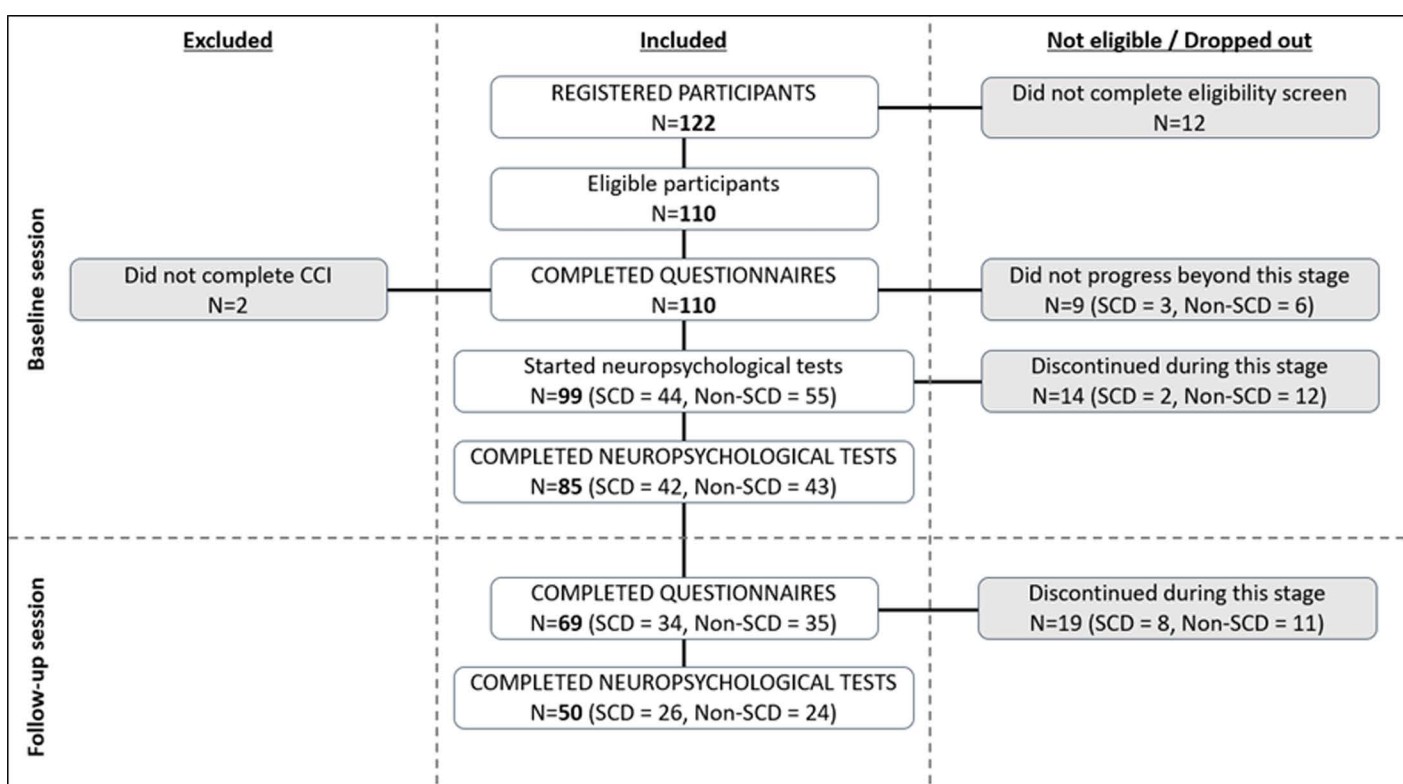

**Fig 1. Participant flow diagram and study completion rates.**

**Table 2. Participant demographics and baseline questionnaire scores.**

| | SCD | N | Non-SCD | N | Test statistic | p |
|---|---|---|---|---|---|---|
| Age, years | 71.00 (12.00) | 47 | 67.00 (8.25) | 60 | $U = 1080.00$ | **0.038** |
| Sex (M/F) | 20/27 | 47 | 20/41 | 61 | $X^2 = 1.09$ | 0.297 |
| Education level | 4.00 (1.00) | 46 | 4.00 (1.00) | 61 | $U = 1687.50$ | 0.061 |
| Confidence using computers | 4.00 (1.00) | 47 | 4.00 (1.00) | 61 | $U = 1553.00$ | 0.424 |
| Sleep (hours) | 7.00 (2.00) | 47 | 7.00 (2.00) | 61 | $U = 1399.00$ | 0.825 |
| Social interaction | 9.00 (3.00) | 47 | 9.00 (3.00) | 61 | $U = 1461.50$ | 0.863 |
| Prev. COVID-19 (Y/N) | 26/21 | 47 | 42/19 | 61 | $X^2 = 2.09$ | 0.149 |
| N diagnosed with long-covid | 1 | 47 | 1 | 61 | $X^2 = 0.04$ | 0.852 |
| Socioeconomic status score | 10.60 (9.10) | 47 | 8.10 (7.10) | 61 | $U = 1321.00$ | 0.481 |
| N diagnosed with dyslexia | 1 | 47 | 1 | 61 | $X^2 = 0.04$ | 0.852 |
| CCI | 23.00 (6.50) | 47 | 15.00 (4.00) | 61 | $U = 0$ | **< 0.001** |
| GDS-15 | 2.00 (3.00) | 47 | 1.00 (2.00) | 61 | $U = 846.00$ | **< 0.001** |
| GAI | 1.00 (4.50) | 47 | 0.00 (2.00) | 61 | $U = 1099.00$ | **0.028** |

Note: data are presented as median (interquartile range) unless otherwise stated. CCI = Cognitive Change Index, GDS-15 = Geriatric Depression Scale-15 item version, GAI = Geriatric Anxiety Inventory.

## Test-retest reliability

Table 3 shows the ICC values for each outcome measure, separated by group (SCD, Non-SCD). Two of the Digit Span Backwards outcome measures ('N correct', and 'Max length') showed moderate test-retest reliability in the SCD group. However, in the Non-SCD group, reliability was poor for all Digit Span Backwards measures. Word Recognition subscores showed poor to good reliability in the Non-SCD group, but poor reliability in the SCD group. ICC values for the Simple Reaction Time task indicated poor reliability across the two groups. Completion time for Trail-Making Test A showed moderate reliability for the Non-SCD group only, whereas for Trail-Making Test B showed moderate reliability in the SCD group only. ICC values for Picture Recognition indicated moderate to good reliability in the Non-SCD group for all measures, and moderate reliability in the SCD group for 'N position correct'. The Sustained Attention to Response Test showed poor to moderate reliability in both groups, for all outcome measures.

## Group differences in online neuropsychological test performance

Eighty-five participants completed the full neuropsychological test battery at baseline (SCD N = 42, Non-SCD = 43; Fig 1). Up to 98 participants provided data for each individual test. The ANCOVA results for group differences in baseline neuropsychological test scores while controlling for age are presented in Table 4. The assumption of homogeneity of regression was tested for each ANCOVA and was non-significant for all. Using the unadjusted alpha level of 0.05, the SCD group scored significantly better than the Non-SCD group for Digit Span Backwards – 'Max length', and Picture Recognition – 'N position correct'. However, these were not significant when using the Bonferroni-adjusted alpha level of 0.003. There were no other significant group differences in baseline neuropsychological test performance.

The ANCOVA results for group differences in baseline-to-follow-up change in scores while controlling for age are presented in Table 5. The assumption of homogeneity of regression was violated for the ANCOVAs of group differences in change scores for the following: Digit Span - N correct, Digit Span - Max Length, Word Recognition - N correct. There were no significant group differences in baseline-to-follow up change in neuropsychological test scores.

**Table 3. Test-retest reliability of NeurOn tests in each group.**

| Measure | SCD | | Non-SCD | |
| --- | --- | --- | --- | --- |
| | ICC | N | ICC | N |
| *Digit Span Backwards* | | | | |
| N correct | **0.73***** | 33 | **0.32*** | 29 |
| N errors | 0.26 | 33 | 0 | 29 |
| Max length | **0.66***** | 33 | 0.11 | 29 |
| *Picture Recognition* | | | | |
| N correct | 0.27 | 32 | **0.71***** | 29 |
| N position correct | **0.50**** | 32 | **0.60***** | 29 |
| False alarms | **0.42**** | 32 | **0.89***** | 29 |
| *Simple Reaction Time* | | | | |
| Average reaction speed | 0.16 | 33 | **0.29*** | 31 |
| N errors | **0.32*** | 33 | 0.07 | 31 |
| *Sustained Attention to Response Test* | | | | |
| N correct | **0.56***** | 29 | **0.47**** | 26 |
| Average reaction speed | **0.70***** | 28 | **0.61***** | 26 |
| N errors | **0.61***** | 29 | **0.63***** | 26 |
| *Trail-Making Test A* | | | | |
| N errors | 0.16 | 26 | 0 | 23 |
| Time to complete | 0.32 | 26 | **0.61***** | 23 |
| *Trail-Making Test B* | | | | |
| N errors | 0 | 26 | 0.40 | 22 |
| Time to complete | **0.74***** | 26 | 0.33 | 22 |
| *Word Recognition* | | | | |
| N correct | **0.37*** | 29 | **0.82***** | 27 |
| N position correct | **0.44**** | 29 | **0.70***** | 27 |
| False alarms | 0.14 | 29 | **0.44**** | 27 |

Note:

*$p < 0.05$,

**$p < 0.01$,

***$p < 0.001$. ICC = Intraclass correlation coefficient.

## Discussion

The aim of the current study was to investigate the test-retest reliability of online, remote neuropsychological assessment in people with and without SCD. Seven online neuropsychological tests were investigated, covering cognitive domains of visual and verbal memory, working memory, attention and psychomotor speed. There was poor to good reliability across all outcome measures. Picture Recognition showed the best evidence for test-retest reliability for the Non-SCD group, with moderate to good reliability for all three outcome measures. Whereas, Digit Span Backwards and the Sustained Attention to Response Test showed good evidence for test-retest reliability for the SCD group, with two and three of the outcome measures showing at least moderate reliability, respectively. We predicted that the tests would show moderate reliability in line with a previous study [27], however the present study used a larger battery with different tests and featured a greater number of outcome measures. Therefore, our results showed greater variability in terms of estimates of reliability. Overall, the best evidence of reliability was found for the Sustained Attention to Response Test and Picture

**Table 4. ANCOVA results for group differences in baseline neuropsychological test scores while controlling for the effect of age.**

| Measure | SCD M (SD) | N | Non-SCD M (SD) | N | F (df) | p | ω² |
|---|---|---|---|---|---|---|---|
| **Digit Span Backwards** | | | | | | | |
| N correct | 6.46 (3.47) | 44 | 5.70 (4.02) | 54 | 2.72 (1, 95) | 0.102 | 0.016 |
| N errors | 3.36 (1.30) | 44 | 2.93 (1.23) | 54 | 3.60 (1, 95) | 0.061 | 0.026 |
| Max length | 4.93 (2.34) | 44 | 4.24 (2.56) | 54 | 4.17 (1, 95) | 0.044 | 0.030 |
| **Picture Recognition** | | | | | | | |
| N correct | 28.21 (2.00) | 43 | 27.20 (4.65) | 54 | 2.01 (1, 94) | 0.160 | 0.010 |
| N position correct | 12.44 (2.72) | 43 | 10.98 (3.70) | 54 | 4.60 (1, 94) | 0.035 | 0.036 |
| False alarms | 0.54 (0.94) | 43 | 1.11 (2.72) | 54 | 1.80 (1, 94) | 0.183 | 0.008 |
| **Simple Reaction Time** | | | | | | | |
| Average reaction speed (ms) | 376.70 (109.42) | 44 | 354.13 (105.68) | 54 | 0.51 (1, 95) | 0.476 | 0.000 |
| N errors | 0.77 (3.33) | 44 | 0.44 (1.14) | 54 | 0.22 (1, 95) | 0.643 | 0.000 |
| **Sustained Attention to Response Test** | | | | | | | |
| N correct | 107.54 (50.39) | 43 | 105.44 (51.14) | 48 | 0.08 (1, 88) | 0.784 | 0.000 |
| Average reaction speed (ms) | 250.24 (122.37) | 43 | 270.39 (140.21) | 48 | 0.65 (1, 88) | 0.422 | 0.000 |
| N errors | 103.84 (50.29) | 43 | 104.35 (56.03) | 48 | 0.29 (1, 88) | 0.590 | 0.000 |
| **Trail-Making Test A** | | | | | | | |
| N errors | 0.88 (1.21) | 42 | 1.61 (4.07) | 44 | 1.40 (1, 83) | 0.240 | 0.005 |
| Time to complete (ms) | 36647.15 (9280.90) | 42 | 32576.49 (10164.89) | 44 | 1.11 (1, 83) | 0.295 | 0.001 |
| **Trail-Making Test B** | | | | | | | |
| N errors | 1.86 (2.95) | 42 | 1.98 (3.58) | 43 | 0.21 (1, 82) | 0.645 | 0.000 |
| Time to complete (ms) | 54022.39 (20969.42) | 42 | 47307.38 (54022.39) | 43 | 0.15 (1, 82) | 0.700 | 0.000 |
| **Word Recognition** | | | | | | | |
| N correct | 24.54 (2.87) | 43 | 24.75 (3.42) | 48 | 0.10 (1, 88) | 0.757 | 0.000 |
| N position correct | 7.40 (3.15) | 43 | 7.04 (3.91) | 48 | 0.73 (1, 88) | 0.396 | 0.000 |
| False alarms | 1.65 (1.79) | 43 | 1.48 (1.75) | 48 | 0.06 (1, 88) | 0.810 | 0.000 |

Note: ms = milliseconds.

Recognition, as these showed moderate to good reliability across both groups for at least one outcome measure. These tests can be recommended for remote and repeated assessment.

A second aim of the study was to explore whether there are group differences (SCD versus Non-SCD) in baseline and longitudinal change in online neuropsychological test scores. At baseline, the SCD group scored significantly better than the Non-SCD group for Digit Span Backwards – 'Max length' (a measure of working memory), and Picture Recognition – 'N position correct' (a measure of spatial memory), which is opposite to what we predicted based on previous research [20]. However, these were not significant when using the Bonferroni-adjusted alpha level which accounts for multiple testing. Therefore, it is possible that these represent false positive results. Additionally, there were no group differences in baseline to follow-up change scores. Given that most of the research into cognition in SCD has employed in-person assessment, it was unclear whether subtle impairment would be detected using online, remote assessment, for which reliability can be impacted by factors specific to this method [21]. It is important to identify reliable online tests as a first step to exploring group differences in performance, and, given the subtle differences reported in the literature to date [20,52], large sample sizes may be required to detect changes when using online assessment methods.

**Table 5. ANCOVA results for group differences in baseline to follow up neuropsychological test change scores while controlling for the effect of age.**

| Measure | SCD M (SD) | N | Non-SCD M (SD) | N | F (df) | p | ω 2 |
|---|---|---|---|---|---|---|---|
| *Digit Span Backwards* | | | | | | | |
| N correct change | 0.55 (2.60) | 33 | 0.97 (5.10) | 29 | 0.58 (1, 59) | 0.451 | 0.000 |
| N errors change | 0.06 (1.48) | 33 | 0.69 (1.71) | 29 | 2.83 (1, 59) | 0.098 | 0.029 |
| Max length change | 0.39 (1.71) | 33 | 0.79 (3.63) | 29 | 0.76 (1, 59) | 0.387 | 0.000 |
| *Picture Recognition* | | | | | | | |
| N correct change | 0.25 (3.02) | 32 | -0.24 (2.67) | 29 | 0.24 (1, 58) | 0.626 | 0.000 |
| N position correct change | 0.41 (2.39) | 32 | 0.07 (2.92) | 29 | 0.09 (1, 58) | 0.763 | 0.000 |
| False alarms change | 0.28 (1.96) | 32 | -0.10 (1.40) | 29 | 0.90 (1, 58) | 0.347 | 0.000 |
| *Simple Reaction Time* | | | | | | | |
| Average reaction speed change (ms) | 46.11 (342.15) | 33 | 36.75 (96.95) | 31 | 0.00 (1, 61) | 0.994 | 0.000 |
| N errors change | 0.06 (0.75) | 33 | 1.48 (7.95) | 31 | 1.25 (1, 61) | 0.269 | 0.004 |
| *Sustained Attention to Response Test* | | | | | | | |
| N correct change | -15.83 (44.67) | 29 | 8.85 (45.97) | 26 | 3.60 (1, 52) | 0.063 | 0.046 |
| Reaction speed change (ms) | 11.26 (119.85) | 29 | -36.06 (105.04) | 26 | 3.33 (1, 52) | 0.074 | 0.040 |
| N errors change | 15.76 (43.59) | 29 | -5.96 (44.67) | 26 | 3.11 (1, 52) | 0.084 | 0.038 |
| *Trail-Making Test A* | | | | | | | |
| N errors change | 1.08 (7.01) | 26 | 0.48 (1.97) | 23 | 0.01 (1, 46) | 0.939 | 0.000 |
| Time to complete change (ms) | 1991.05 (23313.84) | 26 | 206.29 (5796.72) | 23 | 0.00 (1, 46) | 0.971 | 0.000 |
| *Trail-Making Test B* | | | | | | | |
| N errors change | -0.12 (6.02) | 26 | -0.59 (4.04) | 22 | 0.25 (1, 45) | 0.620 | 0.000 |
| Time to complete change (ms) | -1170.22 (17231.23) | 26 | 3008.57 (17220.60) | 22 | 1.63 (1, 45) | 0.208 | 0.013 |
| *Word Recognition* | | | | | | | |
| N correct change | 0.17 (4.12) | 29 | -0.63 (2.27) | 27 | 0.05 (1, 53) | 0.832 | 0.000 |
| N position correct change | 0.07 (3.38) | 29 | -0.82 (3.14) | 27 | 0.26 (1, 53) | 0.613 | 0.000 |
| False alarms change | 0.24 (2.34) | 29 | 0.56 (1.67) | 27 | 0.01 (1, 53) | 0.915 | 0.000 |

Note: ms = milliseconds.

Our results suggest that NeurOn online neuropsychological tests have moderate test-retest reliability in people with SCD and Non-SCD, in particular the Sustained Attention to Response Test and Picture Recognition. In-person equivalents of these tests have shown test-retest reliability estimates of 0.76 (one week follow-up [53]), and 0.60 (one-month follow-up, visual memory [54]), respectively, in healthy control populations. Therefore, the online versions of these tests show comparable reliability in this population of healthy older adults when completed remotely. This suggests that online, remote, completion of these tests is a reliable method for monitoring changes in cognition in this population.

## Limitations

There are a number of limitations to the present study. Some participants discontinued the baseline or follow-up neuropsychological testing sessions and, therefore, there were missing data for the tests. This may have been due to the fully online, remote methodology (i.e., due to lack of additional instruction). Group sizes differed across neuropsychological tests for this reason. However, since the aim of the present research is to understand the feasibility of this methodology for research and clinical practice, this is likely an inevitable consequence of this study design. Future research should investigate whether the rate of non-completion during online, remote assessment paradigms is above that seen in studies using in-person/

supervised assessment methods. Reasons for non-completion were unclear unless participants contacted the lead researcher directly. Therefore, it is not possible to draw firm conclusions about factors contributing to discontinuation of testing in the present study.

There was no option to 'skip' a neuropsychological test during the testing session, meaning that if people encountered technical issues they would be unable to complete the later tasks. This may have reduced the sample sizes for neuropsychological tests towards the end of the battery.

Our definition of SCD was based on the recommended cut-off score on a validated questionnaire (the CCI). This is in line with other studies which have defined SCD using the CCI [40]. However, this method may not completely map on to the definition of SCD proposed by the SCD-Initiative working group [55]. There is considerable variability across studies in the methods used to define SCD making it difficult to compare findings [56]. Therefore, it is not clear whether the finding of no group difference in performance between SCD and Non-SCD in the present study reflects differences in the tests used in the current study to those used in a previous study which found subtle impairment in SCD [20], or whether this reflects differences in the criteria used to define SCD across studies. This should be explored further. There is a need to improve consistency across studies in the definition of SCD. This study was conducted fully online, precluding in-person screening of SCD, meaning SCD could not be clarified by a clinician. It will be particularly important to establish the most suitable method of classifying SCD for online studies.

A further limitation was the decision to not re-assess CCI scores at follow up, which were instead used for classification at baseline assessment. Doing this would have allowed us to understand whether the group composition (SCD, Non-SCD) had altered at the follow up. Finally, while the results of the present study show moderate reliability for a subset of the included tests when completed online and remotely these results are not generaliseable to other online neuropsychological test platforms which may differ in ways to the tests assessed in the current study.

## Conclusion

We found moderate test-retest reliability for two NeurOn tests of memory and attention in people with and without SCD. This suggests online, remote neuropsychological assessment may be a promising option for assessing and monitoring SCD, offering a cheaper alternative to in-person assessment and potentially increasing accessibility for some people. However, our results highlight the need to assess reliability of tests, and their sensitivity to changes associated with SCD to establish the most suitable tests for this purpose. While there are practical issues to be resolved in future research, including exploring issues relating to drop-out, online and remote neuropsychological assessment has the potential to improve efficiency and accuracy of neuropsychological assessment.

## Acknowledgements

The authors would like to thank Alex Howard, AAH Software Ltd, for his advice and support with setting up the study website and the National Institute for Health Research Join Dementia Research service for support provided during recruitment.

## Author contributions

**Conceptualization:** Katie A. Peterson, Adrian Leddy, Michael Hornberger.

**Data curation:** Katie A. Peterson.

**Formal analysis:** Katie A. Peterson.

**Methodology:** Katie A. Peterson, Adrian Leddy, Michael Hornberger.

**Project administration:** Katie A. Peterson, Adrian Leddy, Michael Hornberger.

**Supervision:** Adrian Leddy, Michael Hornberger.

**Writing – original draft:** Katie A. Peterson.

**Writing – review & editing:** Adrian Leddy, Michael Hornberger.

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
