## [Decision Letter · Decision Letter 0]

25 Nov 2024

PDIG-D-24-00474Reliability of online, remote neuropsychological assessment in people with and without subjective cognitive declinePLOS Digital Health Dear Dr. Hornberger, Thank you for submitting your manuscript to PLOS Digital Health. After careful consideration, we feel that it has merit but does not fully meet PLOS Digital Health's publication criteria as it currently stands. Therefore, we invite you to submit a revised version of the manuscript that addresses the points raised during the review process. Please submit your revised manuscript within 60 days Jan 24 2025 11:59PM. If you will need more time than this to complete your revisions, please reply to this message or contact the journal office at digitalhealth@plos.org. Please include the following items when submitting your revised manuscript:* A rebuttal letter that responds to each point raised by the editor and reviewer(s). You should upload this letter as a separate file labeled 'Response to Reviewers '. This file does not need to include responses to any formatting updates and technical items listed in the 'Journal Requirements' section below.* A marked-up copy of your manuscript that highlights changes made to the original version. You should upload this as a separate file labeled 'Revised Manuscript with Track Changes '.* An unmarked version of your revised paper without tracked changes. You should upload this as a separate file labeled 'Manuscript '. If you would like to make changes to your financial disclosure, competing interests statement, or data availability statement, please make these updates within the submission form at the time of resubmission. Guidelines for resubmitting your figure files are available below the reviewer comments at the end of this letter. We look forward to receiving your revised manuscript. Kind regards, Haleh AyatollahiSection EditorPLOS Digital Health Haleh AyatollahiSection EditorPLOS Digital Health Leo Anthony CeliEditor-in-ChiefPLOS Digital Healthorcid.org/0000-0001-6712-6626 **Additional Editor Comments (if provided):****Reviewers' Comments:** Reviewer's Responses to Questions

**Comments to the Author**

1. Does this manuscript meet PLOS Digital Health’s publication criteria ? Is the manuscript technically sound, and do the data support the conclusions? The manuscript must describe methodologically and ethically rigorous research with conclusions that are appropriately drawn based on the data presented.

Reviewer #1: Yes

Reviewer #2: Partly

Reviewer #3: Partly

2. Has the statistical analysis been performed appropriately and rigorously?

Reviewer #1: I don't know

Reviewer #2: Yes

Reviewer #3: Yes

3. Have the authors made all data underlying the findings in their manuscript fully available (please refer to the Data Availability Statement at the start of the manuscript PDF file)?

Reviewer #1: Yes

Reviewer #2: No

Reviewer #3: No

4. Is the manuscript presented in an intelligible fashion and written in standard English?

Reviewer #1: Yes

Reviewer #2: Yes

Reviewer #3: Yes

5. Review Comments to the Author

Reviewer #1: Thank you for letting me read this important piece of work.

The manuscript is well-written and comprehensively explains the methodology. The study design is appropriate for this research question and hypothesis. The results are well explained however since statistical analysis is not my area of expertise, I hope the editors have sent this for review to those who do.

The limtations have been explicitly stated in the discussion section and the conclusions are modest. Therefore, there are no suggestions for revisions from my side.

I wish the authors best of luck.

Reviewer #2: The aim of the research is interesting, but the methodology is not completely justifiable. For example, the statistical analysis relating to the age differences of the two experimental groups involved could have been avoided by adopting homogeneous groups from this point of view, in this way the results would have been cleaner.

The conclusions are not fully supported by the data, which show inconsistencies that, although not significant, are in contrast with results.

Furthermore, the remote test does not allow to completely control the experimental conditions nor the neuropsychological characteristics of the participants, who must simply be "trusted" without a clinical opinion.

Reviewer #3: The paper investigates the reliability of an online-administered suite of cognitive tests on individuals with and without SCD and whether these test measures and changes in the measures show significant group differences. The paper is written well and statistically sound. However, there are some major concerns which the paper needs to address.

1. The main claim from the paper, which is stated in multiple places, e.g., "The results suggest online, remote neuropsychological assessment is a promising option for assessing and monitoring SCD." from the abstract, is not supported by the study and findings. The results show that of the seven tests, only the SART and picture recognition showed moderate test-retest reliability across groups (the picture recognition task barely making into the moderate category). None of the measures, or their changes, showed significant group differences. Uncorrected, rather the measures showed the opposite effect, with SCD reporting higher cognitive scores. As such, the paper cannot claim the results show any promising results from the suite to assess or monitor SCD. The abstract, intro, and conclusions should be edited accordingly. As it stands currently, the main takeaway from the paper is that from a suite of deployed tests, one or two tests show moderate test re-test validity in SCD/non-SCD population.

2. The re-test is scheduled six months after the baseline. The choice for this gap in the study design is not explained. Why were the CCI scores not obtained in the follow-up? Probably the group composition of SCD/Non-SCD changed at follow-up. What is the impact of such possible change on the pursued analysis?

3. The paper refers to their earlier work in [A] and positions that the current work expands on the suite of considered tests for the analysis. In [A], several tests were deployed in the remote test: Reaction Time task, Trail Making Test -A, Trail Making Test -B, Picture Recognition, Spatial Span Backwards, Go/No-Go test, Fragmented Letters, and Virtual Supermarket Test. The current paper only mentions that limited (four) tests were deployed. Am I missing something? Why were some of the tests mentioned in [A] not included in this work?

Other Issues:

1. In the Introduction, please cite tests when they are first mentioned.

2. In the Introduction, this statement is made: "It is unclear whether online neuropsychological tests, completed remotely and unsupervised, show comparable psychometric properties to the 'gold-standard' in-person pen and paper tests." However, this paper does not address this question.

3. In Methods, I am not sure if mentioning developer's name is any helpful or informative and might be rather distracting for the scientific message that the paper intends to convey. Reference/Citation to existing tools should suffice.

4. In Methods, please avoid very generic and marketing-sounding messages like "Some standardised data are available and new tests are being developed."

5. In Methods, the statement "Test-retest reliability has been evaluated for a selection of the NeurOn tests (Reaction

Time, a Go-No/Go test and the Virtual Supermarket Task)" does not list four tests as is mentioned in the next sentence.

6. In the Discussion, "There was poor to good reliability across all outcome measures." is also very generic and could probably be said for any randomly deployed suites of the test. Some specifics would be helpful.

[A] Morrissey, Sol, Rachel Gillings, and Michael Hornberger. "Feasibility and reliability of online vs in-person cognitive testing in healthy older people." Plos one 19, no. 8 (2024): e0309006.

6. PLOS authors have the option to publish the peer review history of their article (what does this mean? ). If published, this will include your full peer review and any attached files.

**Do you want your identity to be public for this peer review?** For information about this choice, including consent withdrawal, please see our Privacy Policy .

Reviewer #1: No

Reviewer #2: No

Reviewer #3: No

---

## [Decision Letter · Decision Letter 1]

18 Feb 2025

Reliability of online, remote neuropsychological assessment in people with and without subjective cognitive decline

PDIG-D-24-00474R1

Dear Prof. Dr. Hornberger,

We are pleased to inform you that your manuscript 'Reliability of online, remote neuropsychological assessment in people with and without subjective cognitive decline' has been provisionally accepted for publication in PLOS Digital Health.

Best regards,

Haleh Ayatollahi

Section Editor

PLOS Digital Health

**Additional Editor Comments (if provided):**

**Reviewer Comments (if any, and for reference):**

Reviewer's Responses to Questions

**Comments to the Author**

1. If the authors have adequately addressed your comments raised in a previous round of review and you feel that this manuscript is now acceptable for publication, you may indicate that here to bypass the “Comments to the Author” section, enter your conflict of interest statement in the “Confidential to Editor” section, and submit your "Accept" recommendation.

Reviewer #1: All comments have been addressed

Reviewer #2: All comments have been addressed

Reviewer #3: All comments have been addressed

2. Does this manuscript meet PLOS Digital Health’s publication criteria ? Is the manuscript technically sound, and do the data support the conclusions? The manuscript must describe methodologically and ethically rigorous research with conclusions that are appropriately drawn based on the data presented.

Reviewer #1: Yes

Reviewer #2: Yes

Reviewer #3: Yes

3. Has the statistical analysis been performed appropriately and rigorously?

Reviewer #1: N/A

Reviewer #2: Yes

Reviewer #3: Yes

4. Have the authors made all data underlying the findings in their manuscript fully available (please refer to the Data Availability Statement at the start of the manuscript PDF file)?

Reviewer #1: Yes

Reviewer #2: Yes

Reviewer #3: Yes

5. Is the manuscript presented in an intelligible fashion and written in standard English?

Reviewer #1: Yes

Reviewer #2: Yes

Reviewer #3: Yes

6. Review Comments to the Author

Reviewer #1: (No Response)

Reviewer #2: The article is ready to be published.

Reviewer #3: I thank the authors for their responsive revision to my reviews. Though it is unfortunate that the deployed tests showed no sensitivity for SCD detection, these 'negative' findings are still a helpful contribution and contributes to the literature. I appreciate the honesty of results that is now reflected in the revised abstract and conclusion.

7. PLOS authors have the option to publish the peer review history of their article (what does this mean? ). If published, this will include your full peer review and any attached files.

**Do you want your identity to be public for this peer review?** For information about this choice, including consent withdrawal, please see our Privacy Policy .

Reviewer #1: No

Reviewer #2: No

Reviewer #3: No
